# AdaShare: Learning What To Share For Efficient Deep Multi-Task Learning

**Ximeng Sun**[1]    **Rameswar Panda**[2]    **Rogerio Feris**[2]    **Kate Saenko**[1,2]

[1]Boston University, [2]MIT-IBM Watson AI Lab, IBM Research

`{sunxm, saenko}@bu.edu, {rpanda@, rsferis@us.}ibm.com`

## Abstract

Multi-task learning is an open and challenging problem in computer vision. The typical way of conducting multi-task learning with deep neural networks is either through handcrafted schemes that share all initial layers and branch out at an adhoc point, or through separate task-specific networks with an additional feature sharing/fusion mechanism. Unlike existing methods, we propose an adaptive sharing approach, called *AdaShare*, that decides what to share across which tasks to achieve the best recognition accuracy, while taking resource efficiency into account. Specifically, our main idea is to learn the sharing pattern through a task-specific policy that selectively chooses which layers to execute for a given task in the multi-task network. We efficiently optimize the task-specific policy jointly with the network weights, using standard back-propagation. Experiments on several challenging and diverse benchmark datasets with a variable number of tasks well demonstrate the efficacy of our approach over state-of-the-art methods. Project page: `https://cs-people.bu.edu/sunxm/AdaShare/project.html`.

## 1   Introduction

Multi-task learning (MTL) focuses on simultaneously solving multiple related tasks and has attracted much attention in recent years. Compared with single-task learning, it can significantly reduce the training and inference time, while improving generalization performance and prediction accuracy by learning a shared representation across related tasks [7, 56]. However, a fundamental challenge of MTL is *deciding what parameters to share across which tasks* for efficient learning of multiple tasks. Most of the prior works rely on hand-designed architectures, usually composed of shared initial layers, after which all tasks branch out simultaneously at an adhoc point in the network (*hard-parameter sharing*) [23, 29, 43, 5, 26, 12]. However, there is a large number of possible options for tweaking such architectures, in fact, too large to tune an optimal configuration manually, especially for deep neural networks with hundreds or thousands of layers. It is even more difficult when the number of tasks grows and an improper sharing scheme across unrelated tasks may cause negative transfer, a severe problem in multi-task learning [52, 27]. Furthermore, it has been empirically observed that different sharing patterns tend to work best for different task combinations [39].

More recently, we see a shift of paradigm in deep multi-task learning, where a set of task-specific networks are used in combination with feature sharing/fusion for more flexible multi-task learning (*soft-parameter sharing*) [39, 16, 48, 33, 49]. While this line of work has obtained reasonable accuracy on commonly used benchmark datasets, it is not computationally or memory efficient, as the size of the model grows proportionally with respect to the number of tasks.

In this paper, we argue that an optimal MTL algorithm should not only achieve high accuracy on all tasks, but also restrict the number of new network parameters as much as possible as the number of tasks grows. This is extremely important for many resource-limited applications such as autonomous vehicles and mobile platforms that would benefit from multi-task learning. Motivated by this, we

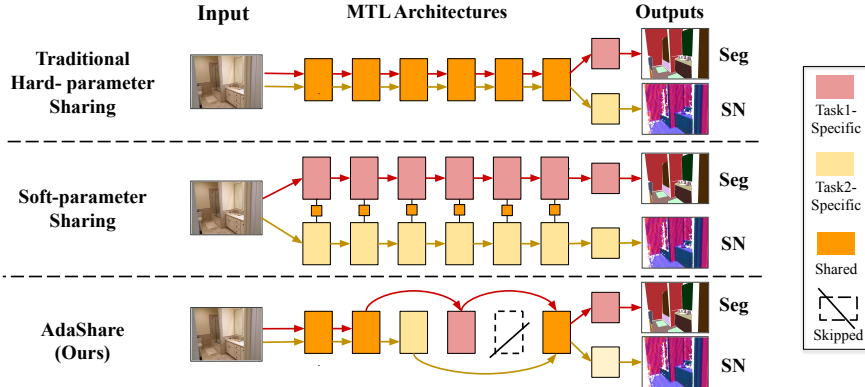

Figure 1: **A conceptual overview of our approach**. Consider a deep multi-task learning scenario with two tasks such as Semantic Segmentation (Seg) and Surface Normal Prediction (SN). Traditional *hard-parameter sharing* uses the same initial layers and splits the network into task-specific branches at an adhoc point (designed manually). On the other hand, *Soft-parameter sharing* shares features via a set of task-specific networks, which does not scale well as the number of tasks increases. In contrast, we propose *AdaShare*, a novel efficient sharing scheme that learns separate execution paths for different tasks through a task-specific policy applied to a single multi-task network. Here, we show an example task-specific policy learned using *AdaShare* for the two tasks.

wish to obtain the best utilization of a single network by exploring efficient knowledge sharing across multiple tasks. Specifically, we ask the following question: *Can we determine which layers in the network should be shared across which tasks and which layers should be task-specific to achieve the best accuracy/memory footprint trade-off for scalable and efficient multi-task learning?*

To this end, we propose *AdaShare*, a novel and differentiable approach for efficient multi-task learning that learns the feature sharing pattern to achieve the best recognition accuracy, while restricting the memory footprint as much as possible. Our main idea is to learn the sharing pattern through a task-specific policy that selectively chooses which layers to execute for a given task in the multi-task network. In other words, we aim to obtain a single network for multi-task learning that supports separate execution paths for different tasks, as illustrated in Figure 1. As decisions to form these task-specific execution paths are discrete and non-differentiable, we rely on Gumbel Softmax Sampling [25, 35] to learn them jointly with the network parameters through standard back-propagation, without using reinforcement learning (RL) [46, 62] or any additional policy network [1, 17]. We design the loss to achieve both competitive performance and resource efficiency required for multi-task learning. Additionally, we also present a simple yet effective training strategy inspired by the idea of curriculum learning [4], to facilitate the joint optimization of task-specific policies and network weights. Our results show that *AdaShare* outperforms state-of-the-art approaches, whilst being more parameter efficient and therefore scaling more elegantly with the number of tasks.

The main **contributions** of our work are as follows:

- We propose a novel and differentiable approach for adaptively determining the feature sharing pattern across multiple tasks (*what layers to share across which tasks*) in deep multi-task learning.

- We learn the feature sharing pattern jointly with the network weights using standard back-propagation through Gumbel Softmax Sampling, making it highly efficient. We also introduce two new loss terms for learning a compact multi-task network with effective knowledge sharing across tasks and a curriculum learning strategy to benefit the optimization.

- We conduct extensive experiments on several MTL benchmarks (NYU v2 [40], CityScapes [11], Tiny-Taskonomy [68], DomainNet [42], and text classification datasets [8]) with variable number of tasks to demonstrate the superiority of our proposed approach over state-of-the-art methods.

## 2 Related Work

**Multi-Task Learning.** Multi-task learning has been studied from multiple perspectives [7, 56, 47]. Early methods have studied feature sharing among tasks using *shallow* classification models [30, 24, 66, 69, 41]. In the context of deep neural networks, it is typically performed with either hard or soft

parameter sharing of hidden layers [47]. *Hard-parameter sharing* usually relies on *hand-designed* architectures composed of hidden layers that are shared across all tasks and specialized branches that learn task-specific features [23, 29, 43, 5, 26, 12]. Only a few methods have attempted to learn multi-branch network architectures, using greedy optimization based on task affinity measures [34, 57] or convolutional filter grouping [6, 54]. In contrast, our approach allows learning of much more flexible architectures beyond tree-like structures, which have proven effective in multi-task learning [38], and relies on a more efficient end-to-end learning method instead of greedy search based on task affinity measures. In parallel, *soft-parameter sharing* approaches, such as Cross-stitch [39], Sluice [48] and NDDR [16], consist of a network column for each task, and define a mechanism for feature sharing between columns. In contrast, our approach achieves superior accuracy while requiring a significantly smaller number of parameters. Attention-based methods, e.g. MTAN [33] and Attentive Single-Tasking [37], introduce a task-specific attention branch per task paired with the shared backbone. Instead of introducing additional attention mechanism, our method adopts adaptive computation that not only encourages positive sharing among tasks via shared blocks but also minimizes negative interference by using task-specific blocks when necessary. More recently, Deep Elastic Network (DEN) [1] specify each network filter to be used or not for each task via learning an additional policy network using complex RL policy gradients [1]. Alternately, we propose a simpler yet effective method which learns to determine the execution of each network layer for each task via direct gradient descent without any additional network. We include a comprehensive comparison with Deep Elastic Network [1] later in our experiments.

**Neural Architecture Search.** Neural Architecture Search (NAS), which aims to automate the design of the network architecture [15], has been studied using different strategies, including reinforcement learning [70, 71], evolutionary computation [53, 45, 44], and gradient-based optimization [61, 32, 65]. Inspired by NAS, in this work we directly learn the sharing pattern in a single network for scalable and efficient multi-task learning. Some recent works [8, 31], in NLP and character recognition, also try to learn the multi-task sharing via RL or evolutionary computation. RL policy gradients are often complex, unwieldy to train and require techniques to reduce variance during training as well as carefully selected reward functions. By contrast, *AdaShare* utilizes a gradient based optimization, which is extremely fast and more computationally efficient than [8, 31].

**Adaptive Computation.** Many adaptive computation methods have been recently proposed to dynamically route information in neural networks with the goal of improving computational efficiency [2, 3, 62, 51, 58, 60, 17, 46, 58, 1]. BlockDrop [62] effectively reduces the inference time by learning to dynamically select which layers to execute per sample during inference, exploiting the fact that ResNets behave like ensembles of relatively shallow networks [59]. Routing networks [46] has also been proposed for adaptive selection of non-linear functions using a recursive policy network trained by reinforcement learning (RL). In transfer learning, SpotTune [17] learns to adaptively route information through finetuned or pre-trained layers. While our approach is inspired by these methods, in this paper we focus on adaptively deciding what layers to share in multi-task learning using an efficient approach that jointly optimizes the network weights and policy distribution parameters, without using RL algorithms [62, 46, 1] or any additional policy network as in [62, 17, 46, 1].

## 3   Proposed Method

Given a set of $K$ tasks $T = \{\mathcal{T}_1, \mathcal{T}_2, \cdots, \mathcal{T}_K\}$ defined over a dataset, our goal is to seek an adaptive feature sharing mechanism that decides what network layers should be shared across which tasks and what layers should be task-specific in order to improve the accuracy, while taking the resource efficiency into account for scalable multi-task learning.

**Approach Overview.** Figure 2 illustrates an overview of our proposed approach. Generally, we seek a binary random variable $\mathbf{u}_{l,k}$ (a.k.a policy) for each layer $l$ and task $\mathcal{T}_k$ that determines whether the $l$-th layer in a deep neural network is selected to execute or skipped when solving $\mathcal{T}_k$ to obtain the optimal sharing pattern, yielding the best overall performance over the task set $T$.

Shortcut connections are widely used in recent network architectures (ResNet [18], ResNeXt [64], and DenseNet [21]) and achieve strong performance in many recognition tasks. These connections make these architectures resilient to removal of layers [59], which benefits our method. In this paper, we consider using ResNets [18] with $L$ residual blocks. In particular, a residual block is said to be shared across two tasks if it is being used by both of them, or task-specific if it is being used by

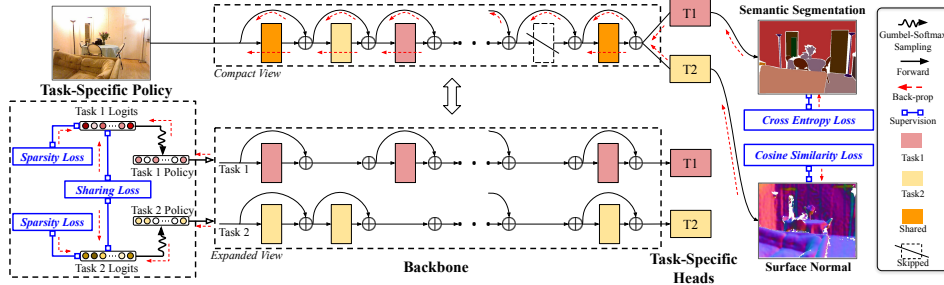

Figure 2: **Illustration of our proposed approach**. *AdaShare* learns the layer sharing pattern among multiple tasks through predicting a select-or-skip policy decision sampled from the learned task-specific policy distribution (logits). These select-or-skip vectors define which blocks should be executed in different tasks. A block is said to be shared across two tasks if it is being used by both of them or task-specific if it is being used by only one task for predicting the output. During training, both policy logits and network parameters are jointly learned using standard back-propagation through Gumbel-Softmax Sampling. We use task-specific losses and policy regularizations (to encourage sparsity and sharing) in training. Best viewed in color.

only one task for predicting the output. In this way, the select-or-skip policy of all blocks and tasks ($\mathbf{U} = \{\mathbf{u}_{l,k}\}_{l \leq L, k \leq K}$) determines the adaptive feature sharing mechanism over the given task set $T$.

As the number of potential configurations for $\mathbf{U}$ is $2^{L \times K}$ which grows exponentially with the number of blocks and tasks, it becomes intractable to manually find such a $\mathbf{U}$ to get the optimal feature sharing pattern in multi-task learning. Instead of handcrafting this policy, we adopt Gumbel-Softmax Sampling [25] to optimize $\mathbf{U}$ jointly with the network parameters $W$ through standard back-propagation. Moreover, we introduce two policy regularizations to achieve effective knowledge sharing in a compact multi-task network, as well as a curriculum learning strategy to stabilize the optimization in the early stages. After the training finishes, we sample the binary decision $u_{l,k}$ for each block $l$ from $\mathbf{u}_{l,k}$ to decide what blocks to select or skip in the task $\mathcal{T}_k$. Specifically, with the help of the select-or-skip decisions, we form a novel and non-trivial network architecture for MTL parameter-sharing, and share knowledge at different levels across all tasks in a flexible and efficient way. At test time, when a novel input is presented to the multi-task network, the optimal policy is followed, selectively choosing what blocks to compute for each task. Our proposed approach not only encourages positive sharing among tasks via shared blocks but also minimizes negative interference by using task-specific blocks when necessary.

**Learning a Task-Specific Policy.** In *AdaShare*, we learn the select-or-skip policy $\mathbf{U}$ and network weights $W$ jointly through standard back-propagation from our designed loss functions. However, each select-or-skip policy $\mathbf{u}_{l,k}$ is discrete and non-differentiable and this makes direct optimization difficult. Therefore, we adopt Gumbel-Softmax Sampling [25] to resolve this non-differentiability and enable direct optimization of the discrete policy $\mathbf{u}_{l,k}$ using back-propagation.

**Gumbel-Softmax Sampling.** The Gumbel-Softmax trick [25, 36] is a simple and effective way to substitutes the original non-differentiable sample from a discrete distribution with a differentiable sample from a corresponding Gumbel-Softmax distribution. We let $\pi_{l,k} = [1 - \alpha_{l,k}, \alpha_{l,k}]$ be the distribution vector of the binary random variable $\mathbf{u}_{l,k}$ that we want to optimize, where the logit $\alpha_{l,k}$ represents the probability that the $l$-th block is selected to execute in the task $\mathcal{T}_k$.

In Gumbel-Softmax Sampling, instead of directly sampling a select-or-skip decision $u_{l,k}$ for the $l$-th block in the task $\mathcal{T}_k$ from its distribution $\pi_{l,k}$, we generate it as,

$$u_{l,k} = \underset{j \in \{0,1\}}{\arg\max} \big( \log \pi_{l,k}(j) + G_{l,k}(j) \big), \tag{1}$$

where $G_{l,k} = -\log(-\log U_{l,k})$ is a standard Gumbel distribution with $U_{l,k}$ sampled from a uniform i.i.d. distribution $\mathrm{Unif}(0, 1)$. To remove the non-differentiable argmax operation in Eq. 1, the Gumbel Softmax trick relaxes one-hot$(u_{l,k}) \in \{0,1\}^2$ (the one-hot encoding of $u_{l,k}$) to $v_{l,k} \in \mathbb{R}^2$ (the soft select-or-skip decision for the $l$-th block in $\mathcal{T}_k$) with the reparameterization trick [25]:

$$v_{l,k}(j) = \frac{\exp\big((\log \pi_{l,k}(j) + G_{l,k}(j))/\tau\big)}{\sum\limits_{i \in \{0,1\}} \exp\big((\log \pi_{l,k}(i) + G_{l,k}(i))/\tau\big)}, \tag{2}$$

where $j \in \{0, 1\}$ and $\tau$ is the temperature of the softmax. Clearly, when $\tau > 0$, the Gumbel-Softmax distribution $p_\tau(v_{l,k})$ is smooth so $\pi_{l,k}$ (or $\alpha_{l,k}$) can be directly optimized by gradient descent, and when $\tau$ approaches 0, the soft decision $v_{l,k}$ becomes the same as one-hot($u_{l,k}$) and the corresponding Gumbel-Softmax distribution $p_\tau(v_{l,k})$ becomes identical to the discrete distribution $\pi_{l,k}$.

Following [17, 61], we optimize the discrete policy $\mathbf{u}_{l,k}, \forall l \leq L, k \leq K$ at once. During the training, we use the soft task-specific decision $v_{l,k}$ given by Eq. 2 in both forward and backward passes [61]. Also, we set $\tau = 5$ as the initial value and gradually anneal it down to 0 during the training, as in [17, 61]. After the learning of the policy distribution, we obtain the discrete task-specific decision $U$ by sampling from the learned policy distribution $p(\mathbf{U})$.

**Loss Functions.** Task-specific losses only optimizes for accuracy without taking efficiency into account. However, we prefer to form a compact sub-model for each single task, in which blocks are omitted as much as possible without deteriorating the prediction accuracy. To this end, we propose a sparsity regularization $\mathcal{L}_{sparsity}$ to enhance the model's compactness by minimizing the log-likelihood of the probability of a block being executed as

$$\mathcal{L}_{sparsity} = \sum_{l \leq L, k \leq K} \log \alpha_{l,k}. \tag{3}$$

Furthermore, we introduce a loss $\mathcal{L}_{sharing}$ that encourages residual block sharing across tasks to avoid the whole network being split up by tasks with little knowledge shared among them. Encouraging sharing reduces the redundancy of knowledge separately kept in task-specific blocks of related tasks and results in an more efficient sharing scheme that better utilizes residual blocks. Specifically, we minimize the weighted sum of $L_1$ distances between the policy logits of different tasks with an emphasis on encouraging the sharing of bottom blocks which contain low-level knowledge. More formally, we define $\mathcal{L}_{sharing}$ as

$$\mathcal{L}_{sharing} = \sum_{k_1, k_2 \leq K} \sum_{l \leq L} \frac{L - l}{L} |\alpha_{l,k_1} - \alpha_{l,k_2}|. \tag{4}$$

Finally, the overall loss $\mathcal{L}$ is defined as

$$\mathcal{L}_{total} = \sum_k \lambda_k \mathcal{L}_k + \lambda_{sp} \mathcal{L}_{sparsity} + \lambda_{sh} \mathcal{L}_{sharing}, \tag{5}$$

where $\mathcal{L}_k$ represent the task-specific losses with task weightings $\lambda_k$. $\lambda_{sp}$ and $\lambda_{sh}$ are the balance parameters for $\mathcal{L}_{sparsity}$ and $\mathcal{L}_{sharing}$ respectively. The additional losses push the policy learning to automatically induce resource efficiency while preserving the recognition accuracy of different tasks.

**Training Strategy.** Following [61, 65], we optimize over the network weights and policy distribution parameters alternately on separate training splits. To encourage the better convergence, we "warm up" the network weights by sharing all blocks across tasks (i.e., hard-parameter sharing) for a few epochs to provide a good starting point for the policy learning. Furthermore, instead of optimizing over the whole decision space in the early training stage, we develop a simple yet effective strategy to gradually enlarge the decision space and form a set of learning tasks from easy to hard, inspired by curriculum learning [4]. Specifically, for the $l$-th ($l < L$) epoch, we only learn the policy distribution of last $l$ blocks. We then gradually learn the distribution parameters of additional blocks as $l$ increases and learn the joint distribution for all blocks after $L$ epochs. After the policy distribution parameters get fully trained, we sample a select-or-skip decision, *i.e.,* feature sharing pattern, from the best policy to form a new network and optimize using the full training set.

**Parameter Complexity.** Note that unlike [8, 17], we optimize over the logits $A = \alpha_{l,k_{l \leq L, k \leq K}}$ for the overall select-or-skip policy $\mathbf{U}$ directly instead of learning a policy network from the semantic task embedding or an image input. As a result, besides the original network, we only occupy $L$ additional parameters for any new task, which results in a negligible parameter count increase over the total number of network parameters. Our model has also a significantly lower number of parameters (about 50% lower while learning two tasks) compared to the recent deep multi-task learning methods [16, 33]. Therefore, in terms of memory, our model scales very well with more tasks learned together.

## 4 Experiments

In this section, we conduct extensive experiments to show that our model outperforms many strong baselines and dramatically reduces the number of parameters and computation for efficient multi-task

Table 1: **NYU v2 2-Task Learning**. *AdaShare* achieves the best performance (bold) on 4 out of 7 metrics and second best (underlined) on 1 metric across Semantic Segmentation and Surface Normal Prediction using less than 1/2 parameters of most baselines. $\mathcal{T}_1$: Semantic Segmentation; $\mathcal{T}_2$: Surface Normal Prediction.

| Model | # Params (%) ↓ | $\mathcal{T}_1$: Semantic Seg. | | | $\mathcal{T}_2$: Surface Normal Prediction | | | | | | | $\Delta_T$ ↑ |
|---|---|---|---|---|---|---|---|---|---|---|---|---|
| | | mIoU ↑ | Pixel Acc ↑ | $\Delta_{\mathcal{T}_1}$ ↑ | Error ↓ | | $\Delta\theta$, within ↑ | | | $\Delta_{\mathcal{T}_2}$ ↑ | |
| | | | | | Mean | Median | 11.25° | 22.5° | 30° | | |
| Single-Task | 0.0 | 27.8 | 58.5 | 0.0 | 17.3 | 14.4 | 37.2 | **73.7** | **85.1** | 0.0 | 0.0 |
| Multi-Task | **- 50.0** | 22.6 | 55.0 | - 12.3 | 16.9 | 13.7 | 41.0 | 73.1 | 84.3 | + 3.1 | - 4.6 |
| Cross-Stitch | 0.0 | 25.3 | 57.4 | - 5.4 | 16.6 | 13.2 | 43.7 | 72.4 | 83.8 | + 5.3 | - 0.1 |
| Sluice | 0.0 | 26.6 | 59.1 | - 1.6 | 16.6 | 13.0 | 44.1 | 73.0 | 83.9 | + 6.0 | + 2.2 |
| NDDR-CNN | + 6.5 | 28.2 | 60.1 | + 2.1 | 16.8 | 13.5 | 42.8 | 72.1 | 83.7 | + 4.1 | + 3.1 |
| MTAN | + 23.5 | 29.5 | 60.8 | + 5.0 | **16.5** | 13.2 | 44.1 | 72.8 | 83.7 | + 5.7 | + 5.4 |
| DEN | - 39.0 | 26.3 | 58.8 | - 2.4 | 17.0 | 14.3 | 39.5 | 72.2 | 84.7 | - 1.2 | - 0.6 |
| *AdaShare* | **- 50.0** | **29.6** | **61.3** | **+ 5.6** | 16.6 | **12.9** | **45.0** | 72.1 | 83.2 | **+ 6.2** | **+ 5.9** |

learning (Tables 1-4). Interestingly, we discover that unlike hard-parameter sharing models, our learned policy often prefers to have task-specific blocks in ResNet's `conv3_x` layers rather than the last few layers (Figure 3: (a)). Moreover, we also show that reasonable task correlation can be obtained from our learned task-specific policy logits (Figure 3: (b), Figure 4).

**Datasets and Tasks.** We evaluate the performance of our approach using several standard datasets, namely **NYU v2** [40] (used for joint Semantic Segmentation and Surface Normal Prediction as in [39, 16], as well as these two tasks together with Depth Prediction as in [33]), **CityScapes** [11], considering joint Semantic Segmentation [9, 55, 20, 19] and Depth Prediction as in [33], and **Tiny-Taskonomy** [68], with 5 sampled representative tasks (Semantic Segmentation, Surface Normal Prediction, Depth Prediction, Keypoint Detection and Edge Detection) as in [52]. We also test *AdaShare* via performing the same task in different data domains such as image classification on 6 domains in **DomainNet** [42] and text classification on 10 publicly available datasets from [8]. More details on the datasets and tasks are included in the supplementary material.

**Baselines.** We compare our approach with following baselines. First, we consider a **Single-Task** baseline, where we train each task separately using a task-specific backbone and a task-specific head for each task. Second, we use a popular **Multi-Task** baseline, in which all tasks share the backbone network but have separate task-specific heads at the end. Finally, we compare our method with state-of-the-art multi-task learning approaches, including **Cross-Stitch Networks** [39] (CVPR'16), **Sluice Networks** [48] (AAAI'19), and **NDDR-CNN** [16] (CVPR'19), which adopt several feature fusion layers between task-specific backbones, **MTAN** [33] (CVPR'19), which introduces task-specific attention modules over the shared backbone, as well as **DEN** [1] (ICCV'19), which uses an additional network to learn channel-wise policy for each task with RL. We use the same backbone and task-specific heads for all methods (including our proposed approach) for a fair comparison.

**Evaluation Metrics.** In both NYU v2 and CityScapes, Semantic Segmentation is evaluated via mean Intersection over Union (mIoU) and Pixel Accuracy (Pixel Acc). For Surface Normal Prediction, we use mean and median angle distances between the prediction and ground truth of all pixels (the lower the better). We also compute the percentage of pixels whose prediction is within the angles of $11.25°$, $22.5°$ and $30°$ to the ground truth [13] (the higher the better). For Depth Prediction, we compute absolute and relative errors as the evaluation metrics (the lower the better) and measure the relative difference between the prediction and ground truth via the percentage of $\delta = \max\{\frac{y_{pred}}{y_{gt}}, \frac{y_{gt}}{y_{pred}}\}$ within threshold 1.25, $1.25^2$ and $1.25^3$ [14] (the higher the better). In Tiny-Taskonomy, we compute the task-specific loss on test images as the performance measurement for a given task, as in [68, 52]. For image classification and text recognition, we report classification accuracy for each domain/dataset. Instead of reporting the absolute task performance with multiple metrics for each task $\mathcal{T}_i$, we follow [37] and report a single relative performance $\Delta_{\mathcal{T}_i}$ with respect to the **Single-Task** baseline to clearly show the positive/negative transfer in different baselines:

$$\Delta_{\mathcal{T}_i} = \frac{1}{|M|} \sum_{j=0}^{|M|} (-1)^{l_j} (M_{\mathcal{T}_i,j} - M_{STL,j})/M_{STL,j} * 100\%, \quad (6)$$

where $l_j = 1$ if a lower value represents better for the metric $M_j$ and 0 otherwise. Finally, we average over all tasks to get overall performance $\Delta_T = \frac{1}{|T|} \sum_{i=1}^{K} \Delta_{\mathcal{T}_i}$.

Table 2: **CityScapes 2-Task Learning**.
$\mathcal{T}_1$: Semantic Segmentation, $\mathcal{T}_2$: Depth Prediction

| Models | # Params ↓ | $\Delta_{\mathcal{T}_1}$ ↑ | $\Delta_{\mathcal{T}_2}$ ↑ | $\Delta_T$ ↑ |
|---|---|---|---|---|
| Multi-Task | **-50.0** | -3.7 | -0.5 | -2.1 |
| Cross-Stitch | 0 | -0.1 | **+5.8** | **+2.8** |
| Sluice | 0 | -0.8 | +4.0 | +1.6 |
| NDDR-CNN | +3.5 | +1.3 | +3.3 | +2.3 |
| MTAN | -20.5 | +0.5 | +4.8 | +2.7 |
| DEN | -44.0 | -3.1 | -1.6 | -2.4 |
| *AdaShare* | **-50.0** | **+1.8** | +3.8 | **+2.8** |

**Single-Task. Seg** - mIoU: 40.2, PAcc: 74,7; **Depth** - Abs.: 0.017, Rel.: 0.33, $\delta < 1.25, 1.25^2, 1.25^3$: 70.3, 86.3, 93.3.

Table 3: **NYU v2 3-Task Learning**. $\mathcal{T}_1$: Semantic Segmentation, $\mathcal{T}_2$: Surface Normal Pred., $\mathcal{T}_3$: Depth Pred.

| Models | # Params ↓ | $\Delta_{\mathcal{T}_1}$ ↑ | $\Delta_{\mathcal{T}_2}$ ↑ | $\Delta_{\mathcal{T}_3}$ ↑ | $\Delta_T$ ↑ |
|---|---|---|---|---|---|
| Multi-Task | **-66.7** | -7.6 | +7.5 | +5.2 | +1.7 |
| Cross-Stitch | 0.0 | -4.9 | + 4.7 | +1.3 | |
| Sluice | 0.0 | -8.4 | +2.9 | 4.1 | -0.5 |
| NDDR-CNN | +5.0 | -15.0 | +2.9 | -3.5 | -5.2 |
| MTAN | +3.7 | -4.2 | **+8.7** | + 3.8 | +2.7 |
| DEN | -62.7 | -9.9 | +1.7 | -35.2 | -14.5 |
| *AdaShare* | **-66.7** | **+8.8** | +7.9 | **+10.1** | **+8.9** |

**Single-Task. Seg** - mIoU: 27.5, PAcc: 58.9; **SN** Mean: 17.5, Median: 15.2, $\Delta\theta < 11.25°, 22.5°, 30°$: 34.9, 73.3, 85.7; **Depth** - Abs.: 0.62, Rel.: 0.25, $\delta < 1.25, 1.25^2, 1.25^3$: 57.9, 85.8, 95.

Table 4: **Tiny-Taskonomy 5-Task Learning**. $\mathcal{T}_1$: Semantic Segmentation, $\mathcal{T}_2$: Surface Normal Prediction, $\mathcal{T}_3$: Depth Prediction, $\mathcal{T}_4$: Keypoint Estimation, $\mathcal{T}_5$: Edge Estimation.

| Models | # Params ↓ | $\Delta_{\mathcal{T}_1}$ ↑ | $\Delta_{\mathcal{T}_2}$ ↑ | $\Delta_{\mathcal{T}_3}$ ↑ | $\Delta_{\mathcal{T}_4}$ ↑ | $\Delta_{\mathcal{T}_5}$ ↑ | $\Delta_T$ ↑ |
|---|---|---|---|---|---|---|---|
| Multi-Task | **-80.0** | - 3.7 | - 1.6 | - 4.5 | 0.0 | + 4.2 | - 1.1 |
| Cross-Stitch | 0.0 | + 0.9 | - 4.0 | **0.0** | - 1.0 | - 2.4 | - 1.3 |
| Sluice | 0.0 | -3.7 | -1.7 | -9.1 | + 0.5 | + 2.4 | - 2.3 |
| NDDR-CNN | +8.2 | -4.2 | -1.0 | - 4.5 | + 2.0 | + 4.2 | - 0.7 |
| MTAN | -9.8 | -8.0 | - 2.8 | - 4.5 | 0.0 | + 2.8 | - 2.5 |
| DEN | -77.6 | -28.2 | - 3.0 | -22.7 | + 2.5 | + 4.2 | - 9.4 |
| *AdaShare* | **-80.0** | **+ 2.3** | **- 0.7** | - 4.5 | **+ 3.0** | **+ 5.7** | **+ 1.1** |

**Single-Task Learning:** Seg: 0.575; SN: 0.707; Depth: 0.022; Keypoint: 0.197; Edge: 0.212

**Experimental Settings.** We use Deeplab-ResNet [9] with atrous convolution, a popular architecture for pixel-wise prediction tasks, as our backbone and the ASPP [9] architecture as task-specific heads. We adopt ResNet-34 (16 blocks) for most scenarios, and use ResNet-18 (8 blocks) for the simple 2-task scenario on the NYU v2 Dataset. For DomainNet, we use the original ResNet-34 as backbone and adopt VD-CNN [10] for text classification. Following [61], we use Adam [28] to update the policy distribution parameters and SGD to update the network parameters. At the end of the policy training, we sample select-or-skip decisions from the policy distribution to be trained from scratch. Specifically, we sample 8 different network architectures from the learned policy and report the best re-train performance as our result. We use cross-entropy loss for Semantic Segmentation as well as classification tasks, and the inverse of cosine similarity between the normalized prediction and ground truth for Surface Normal Prediction. L1 loss is used for all other tasks. Pre-training depends on tasks and we observe that it improves the overall performance of *AdaShare* by 11.3% in NYUv2 3-Task learning. However, to get rid of the unfairness brought by different pretrained model, we start from scratch for a fair comparison among different methods in all our experiments.

**Quantitative Results.** Table 1-4 show the task performance in four different learning scenarios, namely NYU-v2 2-Task Learning, CityScapes 2-Task Learning, NYU-v2 3-Task Learning and Tiny-Taskonomy 5-Task Learning. We report all metrics and the relative performance of two tasks in NYU-v2 2-Task Learning (see Table 1) and report all metrics of Single-Task Baseline and the relative performance of other methods due to the limited space in other cases (see Table 2-4). We recommend readers to refer to supplementary material for the full comparison of all metrics.

In NYU v2 2-Task Learning, *AdaShare* outperforms all the baselines on 4 metrics out of 7 and achieves the second best on 1 metric (see Table 1). Compared to Single-task, Cross-Stitch, Sluice, and NDDR-CNN, which use separate backbones for each task, our approach obtains superior task performance with less than half of the number of parameters. Moreover, *AdaShare* also outperforms the vanilla Multi-Task baseline and DEN [1], the most competitive approaches in terms of number of parameters, showing that it is able to pick an optimal combination of shared and task-specific knowledge with the same number of network parameters without using any additional policy network.

Similarly, for other learning scenarios (Table 2-4), *AdaShare* significantly outperforms all the baselines on overall relative performance while saving at least 50%, up to 80%, of parameters compared to most of the baselines. *AdaShare* also outperforms the Multi-Task baseline and DEN with similar parameter usage. Specifically, for Semantic Segmentation in NYU-v2 3-Task Learning, we observe that the performance of all the baselines are worse than the Single-Task baseline, showing that knowledge from Surface Normal Prediction and Depth Prediction should be carefully selected in

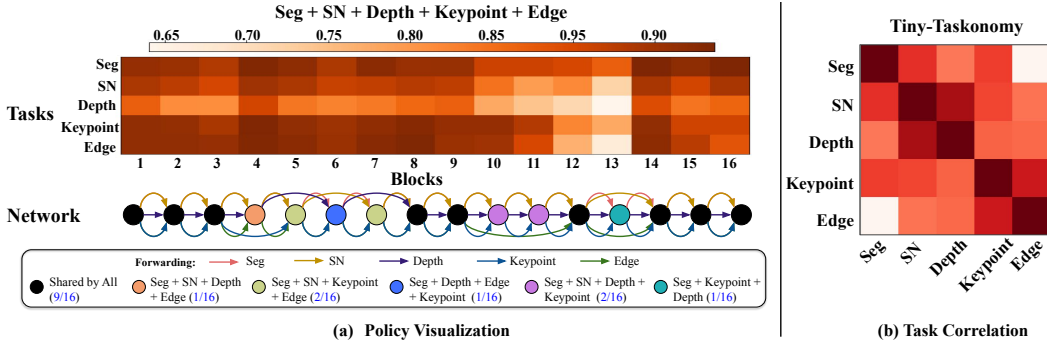

(a) Policy Visualization        (b) Task Correlation

Figure 3: **Policy Visualization and Task Correlation**. (a) We visualize the learned policy logits $A$ in Tiny-Taskonomy 5-Task learning. The darkness of a block represents the probability of that block selected for the given task. We also provide the select-and-skip decision $U$ from our *AdaShare*. In (b), we provide the task correlation, i.e. the cosine similarity between task-specific dataset. Two 3D tasks (Surface Normal Prediction and Depth Prediction) are more correlated and so as two 2D tasks (Keypoint Detection and Edge Detection).

order to improve the performance of Semantic Segmentation. In contrast, our approach is still able to improve the segmentation performance instead of suffering from the negative interference by the other two tasks. The same reduction in negative transfer is also observed in Surface Normal Prediction in Tiny-Taskonomy 5-Task Learning. However, our proposed approach *AdaShare* still performs the best using less than 1/5 parameters of most of the baselines (Table 4).

Moreover, our proposed *AdaShare* also achieves better overall performance across the same task on different domains. For image classification on DomainNet [42], *AdaShare* improves average accuracy over Multi-Task baseline on 6 different visual domains by 4.6% (62.2% vs. 57.6%), with the maximum 16% improvement in *quickdraw* domain. For text classification task, *AdaShare* outperforms the Multi-Task baseline by 7.2% (76.1% vs. 68.9%) in average over 10 different NLP datasets [8] and maximally improves 27.8% in *sogou_news* dataset.

**Policy Visualization and Task Correlation.** In Figure 3: (a), we visualize our learned policy distributions (via logits) and the feature sharing policy in Tiny-Taskonomy 5-Task Learning (more visualizations are included in supplementary material). We also adopt the cosine similarity between task-specific policy logits as an effective representation of task correlations (Figure 3: (b), Figure 4). We have the following key observations. (a) The execution probability of each block for task $k$ shows that not all blocks contribute to the task equally and it allows *AdaShare* to mediate among tasks and decide task-specific blocks adaptive to the given task set. (b) Our learned policy prefers to have more blocks shared only among a sub-group of tasks in ResNet's conv3_x layers, where middle/high-level features, which are more task specific, are starting to get captured. By having blocks shared by a sub-group of tasks, *AdaShare* encourages the positive transfer and relieves the effect of negative transfer, resulting in better overall performance. (c) We clearly observe that Surface Normal Prediction and Depth Prediction, two different 3D tasks, are more correlated, and that Keypoint prediction and Edge detection,

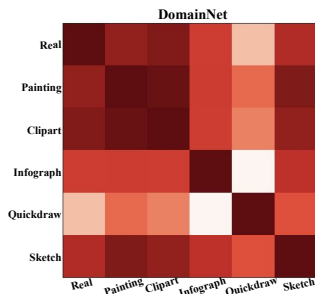

Figure 4: **Task Correlation in DomainNet.** Similar tasks are more correlated, such as *real* is closer to *painting* than *quickdraw*.

two different 2D tasks are more correlated (see Figure 3: (b)). Similarly, Figure 4 shows that the domain *real* is closer to *painting* than *quickdraw* in DomainNet. Both results follow the intuition that similar tasks should have similar execution distribution to share knowledge. Note that the cosine similarity purely measures the correlation between the normalized execution probabilities of different tasks, which is not influenced by the different optimization uncertainty of different tasks.

**Computation Cost (FLOPs).** *AdaShare* requires much less computation (FLOPs) as compared to existing MTL methods. E.g., in Cityscapes 2-task, Cross-stitch/Sluice, NDDR, MTAN, DEN, and AdaShare use 37.06G, 38.32G, 44.31G, 39.18G and 33.35G FLOPs and in NYU v2 3-task, they use 55.59G, 57.21G, 58.43G, 57.71G and 50.13G FLOPs, respectively. Overall, *AdaShare* offers on average about 7.67%-18.71% computational savings compared to state-of-the-art methods over all the tasks while achieving better recognition accuracy with about 50%-80% less parameters.

**Ablation Studies.** We present four groups of ablation studies in NYU-v2 3-Task learning to test our learned policy, the effectiveness of different training losses and optimization method (Table 5).

**Comparison with Stochastic Depth.** Stochastic Depth [22] randomly drops blocks as a regularization during the training and uses the full model in the inference. We compare *AdaShare* with Stochastic Depth in our multi-task setting and observe that *AdaShare* gains more improvement (overall 5.8% improvement in Table 5), which distinguishes *AdaShare* from a regularization technique.

**Comparison with Random Policy.** We perform two different experiments such as 'Random #1' experiment, where we keep the same number of skipped blocks in total for all tasks and randomize their locations and 'Random #2, where we further force the same number of skipped blocks per task as *AdaShare*. We report the best performance among eight samples in each experiment. In Table 5, both random experiments improve the performance of Multi-Task baseline by incorporating shared and task-specific blocks in the model. Also, Random #2 works better than Random #1, which reveals that the number of blocks assigned to each task actually matters and our method makes a good prediction of it. Our model still outperforms Random #2, demonstrating that *AdaShare* correctly predicts the location of those skipped blocks, which forms the final sharing pattern in our approach.

**Ablation on Training Losses and Strategies.** We perform experiments to show the effectiveness of curriculum learning, sparsity regularization $\mathcal{L}_{sparsity}$ and the sharing loss $\mathcal{L}_{sharing}$ in our model. With all the components working, our approach works the best in all three tasks (see Table 5), indicating that three components benefit the policy learning.

**Comparison with Instance-specific Policy.** We employ the same policy network [1] to compute the select-and-skip decision per test image for each task [58, 62]. *AdaShare*, with task-specific policy, outperforms the instance-specific policy (in Table 5), as the discrepancy among tasks dominates over the discrepancy among samples in multi-task learning. Instance-specific methods often introduce extra optimization difficulty and result in worse convergence.

Table 5: **Ablation Study on NYU v2 3-Task Learning**. $\mathcal{T}_1$: Semantic Segmentation, $\mathcal{T}_2$: Surface Normal Prediction, $\mathcal{T}_3$: Depth Prediction.

| Models | $\Delta_{\mathcal{T}_1}\uparrow$ | $\Delta_{\mathcal{T}_2}\uparrow$ | $\Delta_{\mathcal{T}_2}\uparrow$ | $\Delta_T\uparrow$ |
|---|---|---|---|---|
| Stochastic Depth | -2.4 | +7.5 | +4.0 | +3.1 |
| Random # 1 | -2.3 | +5.4 | -0.8 | + 1.3 |
| Random # 2 | + 3.3 | +8.4 | +8.1 | + 6.6 |
| w/o curriculum | +2.1 | +7.4 | +7.2 | + 5.6 |
| w/o $\mathcal{L}_{sparsity}$ | -4.2 | + 4.8 | +1.6 | +0.7 |
| w/o $\mathcal{L}_{sharing}$ | -0.9 | **+9.0** | +8.5 | +5.6 |
| *AdaShare*-Instance | -3.7 | + 5.3 | -22.3 | -6.9 |
| *AdaShare*-RL | -2.8 | 0.0 | -8.2 | -3.7 |
| *AdaShare* | **+8.8** | +7.9 | **+10.1** | **+8.9** |

**Comparison with AdaShare-RL.** We replace Gumbel-Softmax Sampling with REINFORCE to optimize the select-or-skip policy while other parts are unchanged. Table 5 shows *AdaShare* is better than AdaShare-RL in each task and overall performance, in line with the comparison in [63].

**Extension to other Architectures.** We implement *AdaShare* using Wide ResNets (WRN) [67] and MobileNet-v2 [50] in addition to ResNets. *AdaShare* outperforms the Multi-Task baseline by **5.8%** and **3.2%** using WRN and MobileNet respectively in NYU-v2 2-Task (Table 6). We also observe a similar trend on CityScapes 2-Task learning. This shows effectiveness of our proposed approach across different network architectures.

Table 6: **Different Network Architectures on NYU v2 2-Task Learning.** $\mathcal{T}_1$: Semantic Segmentation, $\mathcal{T}_2$: Surface Normal Prediction.

| Models | $\Delta_{\mathcal{T}_1}\uparrow$ | $\Delta_{\mathcal{T}_2}\uparrow$ | $\Delta_T\uparrow$ |
|---|---|---|---|
| **WRN** | | | |
| Multi-Task | -0.35 | 9.63 | 4.64 |
| *AdaShare* | 9.36 | 11.53 | 10.44 |
| **MobileNet-v2** | | | |
| Multi-Task | 0.18 | 8.02 | 4.10 |
| *AdaShare* | 4.16 | 10.61 | 7.39 |

## 5 Conclusion

In this paper, we present a novel approach for adaptively determining the feature sharing strategy across multiple tasks in deep multi-task learning. We learn the feature sharing policy and network weights jointly using standard back-propagation without adding any significant number of parameters. We also introduce two resource-aware regularizations for learning a compact multi-task network with much fewer parameters while achieving the best overall performance across multiple tasks. We show the effectiveness of our proposed approach on five standard datasets, outperforming several competing methods. Moving forward, we would like to explore *AdaShare* using a much higher task-to-layer ratio, which may require increase in network capacity to superimpose all the tasks into a single multi-task network. Moreover, we will extend *AdaShare* for finding a fine-grained channel sharing pattern instead of layer-wise policy across tasks, for more efficient deep multi-task learning.

## Broader Impact

Our research improves the capacity of deep neural networks to solve many tasks at once in a more efficient manner. It enables the use of smaller networks to support more tasks, while performing knowledge transfer between related tasks to improve their accuracy. For example, we showed that our proposed approach can solve five computer vision tasks (semantic segmentation, surface normal prediction, depth prediction, keypoint detection and edge estimation) with 80% fewer parameters while achieving the same performance as the standard approach.

Our approach can thus have a positive impact on applications that require multiple tasks such as computer vision for robotics. Potential applications could be in assistive robots, autonomous navigation, robotic picking and packaging, rescue and emergency robotics and AR/VR systems. Our research can reduce the memory and power consumption of such systems and enable them to be deployed for longer periods of time and become smaller and more agile. The lessened power consumption could have a high impact on the environment as AI systems become more prevalent.

Negative impacts of our research are difficult to predict, however, it shares many of the pitfalls associated with deep learning models. These include susceptibility to adversarial attacks and data poisoning, dataset bias, and lack of interpretablity. Other risks associated with deployment of computer vision systems include privacy violations when images are captured without consent, or used to track individuals for profit, or increased automation resulting in job losses. While we believe that these issues should be mitigated, they are beyond the scope of this paper. Furthermore, we should be cautious of the result of failure of the system which could impact the performance/user experience of the high-level AI systems relied on our research.

## Acknowledgement

This work is supported by DARPA Contract No. FA8750-19-C-1001, NSF and IBM. It reflects the opinions and conclusions of its authors, but not necessarily the funding agents.

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
