[Supplementary Material]

# AdaShare: Learning What To Share For Efficient Deep Multi-Task Learning
# (Supplementary Material)

**Ximeng Sun**[1]    **Rameswar Panda**[2]    **Rogerio Feris**[2]    **Kate Saenko**[1,2]

[1]Boston University, [2]MIT-IBM Watson AI Lab, IBM Research

{sunxm, saenko}@bu.edu, {rpanda@, rsferis@us.}ibm.com

Project Page: https://cs-people.bu.edu/sunxm/AdaShare/project.html

## A   Full Details on the Datasets and Tasks

**CityScapes.** The CityScapes dataset [3] consists of high resolution street-view images. We use this dataset for two tasks: semantic segmentation and depth estimation, as in [7]. We adopt 19-class annotation for semantic segmentation and use the official train/test splits for experiments. During the training, all the input images are resized to 321 x 321 by random flipping, cropping and rescaling and we test on the full resolution 480 x 640.

**NYU v2.** The NYUv2 dataset [9] is consisted with RGB-D indoor scene images. We use this dataset in two different scenarios. First, we consider semantic segmentation and surface normal prediction together [8, 5] and then, use depth prediction along with semantic segmentation and surface normal prediction for experimenting on a 3-task scenario, as in [7]. We use 40-class annotation for semantic segmentation and the official train/val splits which include 795 images for training and 654 images for validation. We use the publicly available surface normals provided by [5] in our experiments. During the training, we resize the input images to 224 x 224 and test on the full resolution 256 x 512.

**Tiny-Taskonomy.** Taskonomy [13] is large-scale dataset consisting of 4.5 million images from over 500 buildings with annotations available for 26 tasks. Considering the huge size of full Taskonomy dataset (~12TB in size), we use its officially released tiny train/val/test splits instead of the full dataset. Tiny Taskonomy consists of 381,840 indoor images from 35 buildings with annotations available for 26 tasks. Following [12], we sampled 5 representative tasks out of 26 tasks for our experiments, namely Semantic Segmentation, Surface Normal Prediction, Depth Prediction, Keypoint Detection and Edge Detection. We use the official train/test splits that include images from 25 buildings for training and images from 5 buildings for testing. This dataset is more challenging as the model has to learn semantic, 3D and 2D structures at the same time for solving these tasks.

**DomainNet.** DomainNet [10] is a recent benchmark for multi-source domain adaptation in object recognition. It is one of the large-scale domain adaptation benchmark with 0.6m images across six domains (clipart, infograph, painting, quickdraw, real, sketch) and 345 categories. We consider each domain as a task and use the official train/test splits in our experiments.

**Text Classification.** We perform text classification on a group of 10 publicly available datasets from [2], namely *ag_news*, *amazon_review_full*, *amazon_review_polarity*, *dbpedia*, *sogou_news*, *yahoo_answers*, *yelp_review_full*, *yelp_review_polarity*, *SST-1* and *SST-2*. These datasets include both multi-class and binary classification tasks. We consider classification within a dataset as task and use the official train/test splits provided by the datasets in our experiments.

Table 1: **Hyper-parameters for NYU v2 2-task learning, CityScapes 2-task learning, NYU v2 3-task learning and Tiny-Taskonomy 5-task learning.** We provide the learning rates (weight lr and policy lr) including $\lambda_{seg}$, $\lambda_{sn}$, $\lambda_{depth}$, $\lambda_{kp}$ and $\lambda_{edge}$ as the task weightings for Semantic Segmentation, Surface Normal Prediction, Depth Prediction, Keypoint Prediction and Edge Detection respectively. $\lambda_{sp}$ and $\lambda_{sh}$ are the weights for sparsity regularization ($\mathcal{L}_{sparsity}$) and sharing encouragement ($\mathcal{L}_{sharing}$) respectively in policy learning.

| Dataset | weight lr | policy lr | $\lambda_{seg}$ | $\lambda_{sn}$ | $\lambda_{depth}$ | $\lambda_{kp}$ | $\lambda_{edge}$ | $\lambda_{sp}$ | $\lambda_{sh}$ |
|---|---|---|---|---|---|---|---|---|---|
| NYU v2 2-task | 0.001 | 0.01 | 1 | 20 | - | - | - | 0.05 | 0.05 |
| CityScapes | 0.0001 | 0.01 | 1 | - | 20 | - | - | 0.01 | 0.1 |
| NYU v2 3-task | 0.001 | 0.01 | 1 | 20 | 3 | - | - | 0.001 | 0.05 |
| Tiny-Taskonomy | 0.001 | 0.01 | 1 | 3 | 2 | 7 | 7 | 0.001 | 0.005 |

Table 2: **CityScapes 2-Task Learning**. Our proposed *AdaShare* achieves the best performance (bold) on five out of seven metrics and second best (underlined) on one metric across Semantic Segmentation and Depth Prediction using less than 1/2 parameters of most baselines.

| Model | # Params ↓ | Semantic Seg. | | Depth Prediction | | | | |
|---|---|---|---|---|---|---|---|---|
| | | mIoU ↑ | Pixel Acc ↑ | Error↓ | | $\delta$, within ↑ | | |
| | | | | Abs | Rel | 1.25 | $1.25^2$ | $1.25^3$ |
| Single-Task | 2 | 40.2 | 74.7 | 0.017 | 0.33 | 70.3 | 86.3 | 93.3 |
| Multi-Task | **1** | 37.7 | 73.8 | 0.018 | 0.34 | 72.4 | 88.3 | 94.2 |
| Cross-Stitch | 2 | 40.3 | 74.3 | **0.015** | **0.30** | 74.2 | 89.3 | **94.9** |
| Sluice | 2 | 39.8 | 74.2 | 0.016 | 0.31 | 73.0 | 88.8 | 94.6 |
| NDDR-CNN | 2.07 | **41.5** | 74.2 | 0.017 | 0.31 | 74.0 | 89.3 | 94.8 |
| MTAN | 2.41 | 40.8 | 74.3 | **0.015** | 0.32 | 75.1 | 89.3 | 94.6 |
| DEN | 1.12 | 38.0 | 74.2 | 0.017 | 0.37 | 72.3 | 87.1 | 93.4 |
| *AdaShare* | **1** | **41.5** | **74.9** | 0.016 | 0.33 | **75.5** | **89.8** | **94.9** |

# B  Implementation Details

Our training is separated into two phases: the Policy Learning Phase and the Re-training Phase. For NYU v2 [4] and CityScapes [3], we update the network 20,000 iterations for both the Policy Learning and Re-training Phases. For Tiny-Taskonomy [13], the network is trained for 100,000 iterations in the Policy Learning Phase and 30,000 in the Re-training Phase. In the Policy Learning Phase, we warm up the network by 20% of total iterations. We train all baselines with the same number of iterations with it in the Re-training Phase to form a fair comparison. In both phases, we use the early stop to get the best performance during the training. In Table 1, we provide the learning rate and loss weightings per dataset. We use the same parameter set for our model and baselines.

# C  Implementation of Baselines

We implement and adapt Cross-Stitch [8], Sluice [11], NDDR-CNN [5], MTAN [7], and DEN [1] to the ResNet architecture following the details in paper and their released code. For Cross-Stitch and Sluice, we insert the linear feature fusion layers after each residual block. For Sluice, we use the orthogonality constraint between two subspaces of the layer-wise feature space [11]. We add each NDDR-layer for feature fusion after each group of blocks, e.g. `conv1_x`, `conv2_x`, as mentioned in [5]. For MTAN, we adapt the attention module which was designed for VGG-16 encoder networks to every residual block in ResNet. In each attention module, we keep the same convolution layers and change input/output channels and spatial dimensions to match the ResNet's architecture [1]. Please refer to [7] for more details. Moreover, instead of 7-class segmentation in [7], we report the standard 19-class segmentation in our work. We also experiment with 7-class segmentation, *AdaShare* achieves average 4% improvement on 5 metrics using 58.5% of parameters fewer than MTAN. For DEN [1], we consult their public code for implementation details and use the same backbone and task-specific heads with *AdaShare* for a fair comparison. We empirically set $\rho = 1$ in DEN to get better performance (compared to $\rho = 0.1$). For Stochastic Depth [6], we randomly drop blocks for each task (with a linear decay rule $p_L = 0.5$ in our implementation) during the training and use all blocks for each task in test.

Table 3: **NYU v2 3-Task Learning**. Our proposed method *AdaShare* achieves the best performance (bold) on ten out of twelve metrics across Semantic Segmentation, Surface Normal Prediction and Depth Prediction using less than 1/3 parameters of most of the baselines.

| Model | # Params ↓ | Semantic Seg. | | Surface Normal Prediction | | | | | Depth Prediction | | | | |
|---|---|---|---|---|---|---|---|---|---|---|---|---|---|
| | | mIoU ↑ | Pixel Acc ↑ | Error ↓ | | $\theta$, within ↑ | | | Error ↓ | | $\delta$, within ↑ | | |
| | | | | Mean | Median | 11.25° | 22.5° | 30° | Abs | Rel | 1.25 | $1.25^2$ | $1.25^3$ |
| Single-Task | 3 | 27.5 | 58.9 | 17.5 | 15.2 | 34.9 | 73.3 | 85.7 | 0.62 | 0.25 | 57.9 | 85.8 | 95.7 |
| Multi-Task | **1** | 24.1 | 57.2 | **16.6** | 13.4 | 42.5 | 73.2 | 84.6 | 0.58 | 0.23 | 62.4 | 88.2 | 96.5 |
| Cross-Stitch | 3 | 25.4 | 57.6 | 17.2 | 14.0 | 41.4 | 70.5 | 82.9 | 0.58 | 0.23 | 61.4 | 88.4 | 95.5 |
| Sluice | 3 | 23.8 | 56.9 | 17.2 | 14.4 | 38.9 | 71.8 | 83.9 | 0.58 | 0.24 | 61.9 | 88.1 | 96.3 |
| NDDR-CNN | 3.15 | 21.6 | 53.9 | 17.1 | 14.5 | 37.4 | **73.7** | 85.6 | 0.66 | 0.26 | 55.7 | 83.7 | 94.8 |
| MTAN | 3.11 | 26.0 | 57.2 | **16.6** | 13.0 | 43.7 | 73.3 | 84.4 | 0.57 | 0.25 | 62.7 | 87.7 | 95.9 |
| DEN | 1.12 | 23.9 | 54.9 | 17.1 | 14.8 | 36.0 | 73.4 | **85.9** | 0.97 | 0.31 | 22.8 | 62.4 | 88.2 |
| *AdaShare* | **1** | **30.2** | **62.4** | **16.6** | **12.9** | **45.0** | 71.7 | 83.0 | **0.55** | **0.20** | **64.5** | **90.5** | **97.8** |

Table 4: **Tiny-Taskonomy 5-Task Learning**. *AdaShare* outperforms the baselines on 3 out of 5 tasks using less than 1/5 parameters of most baselines. While DEN is competitive in terms of number of parameters, *AdaShare* outperforms DEN with an average improvement of 10.5% over all the metrics.

| Models | # Params ↓ | Seg ↓ | SN ↑ | Depth ↓ | Keypoint ↓ | Edge ↓ |
|---|---|---|---|---|---|---|
| Single-Task | 5 | 0.575 | **0.707** | **0.022** | 0.197 | 0.212 |
| Multi-Task | **1** | 0.596 | 0.696 | 0.023 | 0.197 | 0.203 |
| Cross-Stitch | 5 | 0.570 | 0.679 | **0.022** | 0.199 | 0.217 |
| Sluice | 5 | 0.596 | 0.695 | 0.024 | 0.196 | 0.207 |
| NDDR-CNN | 5.41 | 0.599 | 0.700 | 0.023 | 0.196 | 0.203 |
| MTAN | 4.51 | 0.621 | 0.687 | 0.023 | 0.197 | 0.206 |
| DEN | 1.12 | 0.737 | 0.686 | 0.027 | 0.192 | 0.203 |
| *AdaShare* | **1** | **0.562** | 0.702 | 0.023 | **0.191** | **0.200** |

# D  Full Comparison of All Metrics

In this section, we provide the full comparison of all metrics in CityScapes 2-Task Learning, NYU-v2 3-Task Learning and Tiny-Taskonomy 5-Task Learning (see Table 2-4).

# E  FLOPs and Inference Time

We report FLOPs of different multi-task learning baselines and their inference time for all tasks of a single image. Table 5 shows that *AdaShare* reduces FLOPs and inference time in most cases by skipping blocks in some tasks while not adopting any auxiliary networks.

# F  Policy Visualizations

We visualize the policy and sharing patterns learned by *AdaShare* in NYU-v2 2-Task Learning, CityScapes 2-Task Learning and NYU-v2 3-Task Learning (see Figure 1). The observations of policy visualization in the main paper still hold in these scenarios.

We experiment on five tasks (Semantic Segmentation, Surface Normal Prediction, Depth Prediction, Keypoint Prediction and Edge Prediction) for Tiny-Taskonomy dataset. In the main paper (see Section 4.1), we visualize the policy decision for five tasks. In this section, we further investigate the sharing patterns of subset of tasks (see Figure 2), e.g., Semantic Segmentation and Surface Normal

Table 5: **FLOPs and Inference Time Comparison among Cross-stitch, Sluice, NDDR-CNN, MTAN, DEN and *AdaShare***. *AdaShare* consumes fewer FLOPs and shorter inference time in most scenarios.

| Models | GFLOPs | | | | Inference Time (ms) | | | |
|---|---|---|---|---|---|---|---|---|
| | NYU v2 2-Task | CityScapes 2-Task | NYU v2 3-Task | Taskonomy 5-Task | NYU v2 2-Task | CityScapes 2-Task | NYU v2 3-Task | Taskonomy 5-Task |
| Cross-Stitch | 19.71 | 37.06 | 55.59 | 92.64 | 11.48 | 21.29 | 32.36 | 57.64 |
| Sluice | 19.71 | 37.06 | 55.59 | 92.64 | 11.48 | 21.29 | 32.36 | 57.64 |
| NDDR-CNN | 20.98 | 38.32 | 57.21 | 100.55 | 11.14 | 20.21 | 30.63 | 52.34 |
| MTAN | 24.07 | 44.31 | 58.43 | 82.99 | 15.85 | 29.68 | 40.08 | 60.96 |
| DEN | 21.84 | 39.18 | 57.71 | 94.77 | 14.69 | 26.10 | 38.30 | 62.41 |
| *AdaShare* | 18.48 | 33.35 | 50.13 | 87.75 | 10.87 | 19.15 | 28.96 | 51.01 |

Figure 1: **Policy Visualization**. We visualize the learned policy logits $A$ in NYU-v2 2-Task Learning, CityScapes 2-Task Learning and NYU-v2 3-Task Learning. Best viewed in color.

Figure 2: **Policy Visualization of subset of tasks on Tiny-Taskonomy.** We visualize the sharing patterns of subset of tasks: {Semantic Segmentation (Seg), Surface Normal Prediction (SN)}, {Seg, Depth Prediction (Depth)} and {Seg, SN and Depth}. For example, in (a), Seg and SN share 14 out of 16 blocks in total and Seg owns 2 task-specific blocks. Best viewed in color.

Prediction (Figure 2.(a)), Semantic Segmentation and Depth Prediction (Figure 2.(b)) and Semantic Segmentation, Surface Normal Prediction and Depth Prediction (Figure 2.(c)). These subset of tasks are same as the tasks considered in NYU v2 2-Task Learning, CityScapes 2-Task Learning and NYU v2 3-Task Learning respectively. In each subset of tasks, we both have shared blocks and task-specific (or not shared by all tasks) blocks. The sharing patterns help the model to share the knowledge between tasks when necessary and own the individual knowledge for a single task.

## G   Class-wise Segmentation Performance

The performance of Semantic Segmentation can be easily affected by both Surface Normal Prediction and Depth Prediction tasks on NYU v2 dataset, but our method mitigates this negative interference and further improves the performance. In this section, we closely investigate the performance (Pixel Accuracy) per class and their relationship with the number of labeled pixels. From Figure 3, we find that we improve the performance of most classes including those with less labeled data compared to MTAN [7] (the most competitive MTL baseline in semantic segmentation performance).

## H   Qualitative Visualization

In this section, we visualize the results of Multi-Task, MTAN (the best baseline), DEN (ICCV 2019) and *AdaShare* in NYU v2 3-task learning. From the comparison (see Figure 4), we observe that *AdaShare* predicts the class label more accurately in Semantic Segmentation; predicts the normal

Figure 3: **Change in Pixel Accuracy for Semantic Segmentation classes of *AdaShare* over MTAN (blue bars).** The class is ordered by the number of pixel labels (the black line). Compare to MTAN, we improve the performance of most classes including those with less labeled data.

Figure 4: **Qualitative Visualization of Multi-Task, MTAN, DEN and *AdaShare* Performance in NYU v2 3-task Learning.** The red boxes represent the regions of interest. Our proposed method, *AdaShare* gives more accurate prediction and clearer contour in Semantic Segmentation (Seg), Surface Normal Prediction (SN) and Depth Prediction (Depth). Best viewed in color.

vector closer to the ground truth in Surface Normal Prediction; gives clearer contour of object in Semantic Segmentation, Surface Normal Prediction and Depth Prediction.

## I  Ablation Studies on CityScapes

We present four groups of ablation studies in CityScapes 2-Task Learning to test our learned policy, the effectiveness of different training losses and optimization method (Table 6). Similar to ablation studies on NYU-v2 3 task learning (in the main paper), our proposed *AdaShare* outperforms its variants in most of individual metrics and overall performance.

Table 6: **Ablation Studies on CityScapes 2-Task Learning**. $\mathcal{T}_1$: Semantic Segmentation, $\mathcal{T}_2$: Depth Prediction

| Model | $\mathcal{T}_1$: Semantic Seg. | | | $\mathcal{T}_2$: Depth Prediction | | | | | | $\Delta_T \uparrow$ |
|---|---|---|---|---|---|---|---|---|---|---|
| | mIoU ↑ | Pixel Acc ↑ | $\Delta_{\mathcal{T}_1} \uparrow$ | Error ↓ | | $\delta$, within ↑ | | | $\Delta_{\mathcal{T}_2} \uparrow$ | |
| | | | | Abs | Rel | 1.25 | $1.25^2$ | $1.25^3$ | | |
| Stochastic Depth | 41.0 | 74.2 | +0.7 | **0.016** | 0.37 | 71.0 | 86.0 | 92.9 | -1.2 | -0.3 |
| Random # 1 | 40.7 | 74.6 | +0.8 | **0.016** | 0.35 | 74.7 | 88.2 | 94.0 | +1.8 | +1.3 |
| Random # 2 | 41.2 | 74.9 | +1.4 | 0.017 | 0.36 | 74.1 | 88.2 | 93.7 | -0.2 | +0.6 |
| w/o curriculum | 40.4 | 74.8 | -1.0 | 0.017 | **0.33** | 75.1 | 88.9 | 94.5 | 0.0 | -0.5 |
| w/o $\mathcal{L}_{sparsity}$ | 40.8 | 74.8 | +0.8 | **0.016** | 0.34 | 73.8 | 89.2 | 94.7 | +2.5 | +1.7 |
| w/o $\mathcal{L}_{sharing}$ | **41.5** | **74.9** | **+1.8** | **0.016** | 0.35 | 74.0 | 88.7 | 94.4 | +1.8 | +1.8 |
| *AdaShare*-Instance | **41.5** | 74.7 | +1.6 | **0.016** | 0.33 | 74.4 | 89.5 | **94.9** | +3.4 | +2.5 |
| *AdaShare*-RL | 40.2 | 74.4 | -0.2 | 0.018 | 0.36 | 71.7 | 87.4 | 93.7 | -2.3 | -1.2 |
| *AdaShare* | **41.5** | **74.9** | **+1.8** | **0.016** | **0.33** | **75.5** | **89.8** | **94.9** | **+3.8** | **+2.8** |

## J  Full Comparison of Ablation Studies on NYU-v2 3-Task

In addition to the relative performance of ablation studies in NYU-v2 3-Task Learning, we provide the full comparison of all metrics (see Table 7).

Table 7: **Ablation Studies in NYU v2 3-Task Learning**.

| Model | Semantic Seg. | | Surface Normal Prediction | | | | | Depth Prediction | | | | |
|---|---|---|---|---|---|---|---|---|---|---|---|---|
| | mIoU ↑ | Pixel Acc ↑ | Error ↓ | | $\theta$, within ↑ | | | Error ↓ | | $\delta$, within ↑ | | |
| | | | Mean | Median | 11.25° | 22.5° | 30° | Abs | Rel | 1.25 | $1.25^2$ | $1.25^3$ |
| Stochastic Depth | 26.4 | 58.5 | 16.6 | 13.4 | 42.6 | **72.9** | **84.7** | 0.60 | 0.22 | 58.8 | 87.7 | 96.9 |
| Random #1 | 26.4 | 59.7 | 16.9 | 13.9 | 41.5 | 71.6 | 84.2 | 0.61 | 0.23 | 59.0 | 87.4 | 96.7 |
| Random #2 | 28.4 | 60.9 | 16.7 | 13.0 | 44.7 | 71.9 | 83.0 | 0.56 | 0.21 | 62.7 | 89.7 | 97.6 |
| w/o curriculum | 27.9 | 60.5 | 17.0 | 13.1 | 44.1 | 71.4 | 82.6 | 0.58 | 0.21 | 63.0 | 89.0 | 96.8 |
| w/o $\mathcal{L}_{sparsity}$ | 25.8 | 57.6 | 16.9 | 14.0 | 41.6 | 70.9 | 83.6 | 0.63 | 0.23 | 58.0 | 86.3 | 96.5 |
| w/o $\mathcal{L}_{sharing}$ | 26.6 | 59.8 | **16.5** | **12.9** | 44.8 | 72.3 | 83.4 | 0.56 | 0.21 | 64.0 | 89.7 | 97.4 |
| *AdaShare*-Instance | 27.3 | 55.0 | 17.0 | 13.8 | 41.3 | 72.1 | 83.4 | 0.56 | 0.21 | 64.0 | 89.7 | 90.6 |
| *AdaShare*-RL | 26.9 | 56.9 | 18.3 | 14.7 | 39.0 | 69.0 | 81.7 | 0.74 | 0.27 | 51.7 | 83.3 | 95.7 |
| *AdaShare* | **30.2** | **62.4** | 16.6 | **12.9** | **45.0** | 71.7 | 83.0 | **0.55** | **0.20** | **64.5** | **90.5** | **97.8** |

## Footnotes

[1]Note that it would cause the difference in the number of parameters.