[Reviews · NeurIPS 2020]

Review 1

Summary and Contributions: This work proposes to search feature sharing strategy for multi-task learning. It relies on standard back-propagation to jointly learn feature sharing policy and network weights. It also uses two regularizations for learning a compact network. Experiments are conducted on three standard MTL datasets and improved results are obtained.

Strengths: + Searching feature sharing strategy for multi-task learning is interesting. + The related work is appropriately discussed and compared. + The performance looks good compared to other methods in MTL benchmarks. + Detailed ablation studies are also presented.

Weaknesses: - The method highly relies on the ResNet residual block for search. How does this method extend to other network architecture (e.g., MobileNet or Inception)? - The motivation of sparsity loss is enhancing the compact of model. It is not clear that such strategy can largly affect the multi-task learning performance accroding to Table 5. - RL-based methods are widely used in many existing NAS methods. It is also not clear to me why RL-based searching method performs much worse than the proposed differentiable-based method in Table 5. - I suggest that computation cost (FLOPs) can be presented other than #params to show the efficiency of MTL methods.

Correctness: Yes

Clarity: Yes

Relation to Prior Work: Yes

Reproducibility: Yes

Additional Feedback: See above Update after the rebuttal: I read the rebuttal and agree that this paper can be accepted.


Review 2

Summary and Contributions: In this paper, they introduce a multi-task scheme in which they adaptively select what to share across various tasks. That is, they want to decide which layers could be shared across specific tasks, and which layers are the best to be task-specific. They learn this policy sharing parameters while learning the model parameters. Their method outperformed several existing MTL on three multi-task learning benchmarks.

Strengths: - Paper is well-written and well-organized. I liked figures 1 and 2, which compare their method with 2 popular MLT scheme as well as depicting their methods with more details. - They introduced a MTL method in which they learn how tasks are correlated with each other and which layers are the best for which tasks during training. The cost is learning more parameters (policy-sharing) which does not introduce much computational overhead. This is a very interesting innovation as we can learn which tasks are related after training the model. (They explained in Policy Visualization and Task Correlation section how tasks are related in their experiments) - Thorough and extensive experiments, especially their ablation study where they examine the effect of each loss function. I also like their qualitative analysis in figure 3, where they show how task relatedness are learned after training.

Weaknesses: - I am wondering how this framework is easy and straightforward to apply for MTL tasks. I did not see discussion on the paper (or am I missing that?) and also any discussion about the computational cost? - Even though their method works well on three MTL benchmarks, I would recommend adding one/two datasets in another domain such as language.

Correctness: yes

Clarity: yes

Relation to Prior Work: yes

Reproducibility: Yes

Additional Feedback: I read the rebuttal, I like the paper, my score is the same.


Review 3

Summary and Contributions: This paper explores the layer dropping/pruning technique in multi-task learning (MTL). It proposes to train a layer-drop policy for each task individually. Through this, each task may choose to skip or execute each layer and in theory the network can learn what to share and what not to share. Overall, this strategy has been shown effective on MTL with a few tasks (<= 5).

Strengths: In general, the paper is easy to follow. It also contains a good amount of experiments that clearly shows its improvement over baselines. It demonstrates that dropping layers for individual tasks in MTL may mitigate negative transfer.

Weaknesses: Despite its effectiveness, there are certain aspects that I am concerned about, as following: (1) The idea of layer dropping is not new. It has been explored for regularization [1] as well as structured pruning [2, 3]. In addition, methods of routing subnetwork and learning task-specific params for each task [4] have also been studied before. (2) The three datasets included are similar in nature. It would be more comprehensive to additionally test on a dataset with larger numbers of tasks. (3) It is not very clear what are sources of improvement for this method and if it remains effective for larger datasets. First, I think that the method’s main advantage is not memory efficiency since it is at least as large as a standard MTL network (denoted as Multi-Task in the paper). Therefore, the main point should be addressing why it has been so effective. Overall, the results seem to show that there is negative transfer [5] among tasks and thus dropping some layers may benefit certain tasks. This is particularly true given that ‘Random #2’ performs well. Then some unanswered questions include: (a) How is the number dropped for each task correlated to performance gain? (b) How does the method compare with adding task-specific adapter layers? (c) How does it compare to learning a stochastic depth pattern for each task (using expected value instead of pruning)? (d) What would happen if we use a higher task-to-layer ratio (20 tasks v.s. 18 layers, etc)? (e) How would pre-training play a role here? Will it improve or degenerate AdaShare? (f) What about other components (e.g. channels)? Can we do something similar? [1] Deep networks with stochastic depth. ECCV 2016. [2] Reducing transformer depth on demand with structured dropout. ICLR 2020. [3] Data-driven sparse structure selection for deep neural networks. ECCV 2018. [4] Adversarial examples improve image recognition. CVPR 2020. [5] Characterizing and avoiding negative transfer. CVPR 2019.

Correctness: The method is generally correct.

Clarity: The paper is well written with good visualizations.

Relation to Prior Work: Some citations are missing (see above). I recommend adding some discussion on how this work differs from task-specific stochastic depth and structured pruning, and its relation to addressing negative transfer.

Reproducibility: Yes

Additional Feedback:


Review 4

Summary and Contributions: The paper introduces a method for multi task learning with an adaptive sharing approach called adashare. This method is different from current soft parameter sharing methods as it takes resource efficiency into consideration instead of learning a task specific network per task. It differs from hard parameter sharing as well as it has no hand set network split for each task. The paper uses Gumbel softmax sampling to jointly learn the execution paths per task. The contributions are a differentiable approach for adaptive feature sharing without requiring any reinforcement learning, two new loss terms to balance learning new features vs sharing features, and ablation experiments on some well known datasets.

Strengths: The method is very interesting. It is simple and clear and makes sense. I like the emphasis on efficiency. This seems like a nice method that has advantages of both some soft-parameter sharing methods and some hard-parameter sharing methods by being adaptable but also efficient in memory and computation. I especially like the paragraph on line 28 about more flexible multi-task learning but it’s important to be computationally and memory efficient. I also think the added loss terms (sparsity loss and sharing loss) are a nice addition in Equations 3 and 4. They make a lot of sense to have and I haven’t seen their exact like before in this type of work. I like Figure 3 and the interesting relationships it shows. I think that’s an excellent addition to the paper and worth having for others to note/cite in future works. Despite the weaknesses of only one architecture, only related tasks, and some small clarity issues, I think the paper is well-written and I think the concept is novel and clever. I would put my score at 7.5 so I’m rounding up to an 8.

Weaknesses: The method learns a policy distribution for a feature sharing pattern but then is optimized on the full training set. It’s not clear the effect of this. It would be simpler and cleaner if the network didn’t need several stages of training. The curriculum learning method does make sense but nowhere in the paper is an ablation study showing this except for the line in Table 5 which is only one experiment. It doesn’t seem like the curriculum learning is that large of an effect on this specific example. Furthermore, I don’t believe the random policies are that meaningful of a comparison in Table 5. Random is a very low baseline. Its unusual that Random #2 does so well. The biggest weakness I see is that this is only shown for one network architecture and only very related tasks. It would be interesting to see some more diverse and different tasks. FIg3B at least shows that some of these are more related than others so perhaps an ablation study showing the architectures found for the 2 closest tasks compared to the 2 furthest tasks. I think that would be interesting to show that the closer tasks have more shared features. In terms of only one architecture, the paper only shows a large resnet style encoder which is interesting since the paper argues for more memory efficiency. It would be good to show this also works on a smaller style architecture like mobilenetv3, etc.

Correctness: The type of surface normals for NYUv2 are not shown or cited appropriately. There are several different surface normals produced for this dataset. Which did the authors use? (Ladicky, Eigen, Hickson, etc. all have different surface normals produced for this dataset)

Clarity: Line 23-24: Cumbersome and hard to read sentence. Line 80-81: The method attempts to minimize negative interference. None of the experiments prove that it does this. I see the theory and argument but I think the language is important here. Nothing is shown that proves this. Line 154 introduces P_T but doesn’t define it. Perhaps its better to define that previously in equation 1? Line 153, remove Clearly. It’s understandable but I wouldn’t call that clear. Best to avoid language like that and “it’s trivial that…”, etc. Following that, in line 156, there is no variable that decides where to split the distribution to one-hot during training. Is it greater than probability 0.5 is 1 and less than 0? This is not made clear in the paper. Line 207-210, why are the citations in a different format with conference/year? In Equation 6, what is STL? Table 3, Sluice/T3 is missing a +/-.

Relation to Prior Work: This paper has a very similar title (Adashare) to: Kong, Linghe, et al. "AdaSharing: Adaptive data sharing in collaborative robots." IEEE Transactions on Industrial Electronics 64.12 (2017): 9569-9579. It’s slightly different but at least something to consider in case the authors decide to change the name to be more distinct. Otherwise the relationship to previous work is well established.

Reproducibility: Yes

Additional Feedback: Many of the cifations have arxiv versions cited instead of the correct published versions. Please take the time to correct these to the published versions. Google scholar sometimes has these wrong. Ex: 1] Ahn, Chanho, Eunwoo Kim, and Songhwai Oh. "Deep elastic networks with model selection for multi-task learning." Proceedings of the IEEE International Conference on Computer Vision. 2019.

[Author Response · NeurIPS 2020]

We thank reviewers for their feedback. As stated by reviewers, the work is novel (**R4**), very interesting (**R1**, **R2**, **R4**),
backed with extensive experiments (**R2**, **R3**), detailed ablation studies (**R1**, **R2**, **R4**) and qualitative analysis (**R2**, **R4**).

**R2**, **R3**, **R4**: **Experiments on Additional Datasets.** We experiment on more di-
verse DomainNet [a] with 6 dissimilar image classification tasks using ResNet34
and text recognition with larger number of NLP tasks (10 different publicly
available datasets from [b]) using VD-CNN [c]. AdaShare improves average
accuracy over 'Multi-Task' by **4.6%** (max. **16%** in *quickdraw*) for DomainNet,
and **7.2%** (max. **27.8%** in *sogou_news*) for text recognition. Similar to Fig.3.b,
we visualize task relationship on DomainNet, which shows similar tasks are
more correlated, such as *real* is closer to *painting* than *quickdraw* (Fig. 1).

| Models | $\Delta_{\mathcal{T}_1} \uparrow$ | $\Delta_{\mathcal{T}_2} \uparrow$ | $\Delta_{\mathcal{T}} \uparrow$ |
|---|---|---|---|
| **WRN** | | | |
| Multi-Task | -0.35 | 9.63 | 4.64 |
| AdaShare | 9.36 | 11.53 | 10.44 |
| **MobileNet-v2** | | | |
| Multi-Task | 0.18 | 8.02 | 4.10 |
| AdaShare | 4.16 | 10.61 | 7.39 |

Table 1: NYU v2 2-Task. $\mathcal{T}_1$: Semantic Seg., $\mathcal{T}_2$: Surface Normal Pred.

**R1**, **R4**: **Extension to other Architectures.** We implemented AdaShare using
Wide ResNets (WRN) and MobileNet-v2 in addition to ResNets. AdaShare outperforms 'Multi-Task' by **5.8%** and
**3.2%** using WRN and MobileNet respectively in NYU v2 2-Task (Tab. 1). We observe a similar trend on CityScapes.

**R1**, **R2**: **Computation Cost (FLOPs).** AdaShare requires much less computation (FLOPs) as compared to existing
MTL methods. E.g., in Cityscapes 2-task, Cross-stitch/Sluice, NDDR, MTAN, DEN, and AdaShare use 37.06G, 38.32G,
44.31G, 39.18G and **33.35G** FLOPs and in NYU v2 3-task, they use 55.59G, 57.21G, 58.43G, 57.71G and **50.13G**
FLOPs, respectively. Overall, AdaShare offers on average about **7.67%-18.71%** computational savings compared to
SOTA methods over all the tasks while achieving better recognition accuracy with about 50%-80% less parameters.

**R1**: **Sparsity Loss.** Sparsity loss enhances compactness and also helps learning task-specific
layers (i.e., skipped layers in one task form the task-specific layers of other tasks) which
potentially reduces negative transfer, leading to performance improvement in MTL.

**R1**: **RL-based Methods.** Table 5 shows that AdaShare is better than AdaShare-RL, in line
with comparison in [57]. This is due to RL policy gradients are often complex, unwieldy to
train and require techniques to reduce variance during training. In contrast, Gumbel Softmax
sampling (used in this work) makes the framework fully differentiable with more efficient
gradient feedback from the training loss and also easier to optimize, as shown in [29,55,59].

Figure 1: Task Correlations in DomainNet.

**R2**: **Applications.** Our approach is easy and straightforward to apply: during training, we
learn an feature sharing pattern and then at testing, the learned pattern is followed, selectively
choosing what layers to compute for each task. Our source code will be publicly available (also included in supp).

**R3**: **Difference from Prior Works.** While methods in [1-4] enhance efficiency of a single classification task via
training regularization, AdaShare **jointly** learns feature sharing patterns among multiple tasks via adaptive computation.
Compared to *task-specific residual adapters*, AdaShare requires **36.2%** less parameters and **23.4%** less FLOPs, with
an overall improvement of **5.6%** on NYU-v2 3-Task learning. As suggested, we also compare with *task-specific*
*stochastic depth* and find that AdaShare outperforms it by **5.7%** on NYU-v2 3-Task. Our approach is effective as it
not only encourages positive sharing among tasks via shared blocks but also minimizes negative interference by using
task-specific blocks when necessary. Thanks for the references–we will cite them in our final version.

**R3**: **Dropped Blocks vs Performance.** The average probability of a block to be dropped depends on the real task
difficulty and hence more blocks can be dropped for an easier task without affecting the performance. AdaShare
mediates among tasks and automatically decides shared and task-specific blocks adaptive to given task set.

**R3**: **Higher Task-to-Layer Ratio.** We believe using a much higher task-to-layer ratio may require increase in network
capacity to superimpose all the tasks into a single multi-task network. AdaShare can be extended to dynamically grow
the network capacity in addition to feature sharing, which is an interesting topic for future work.

**R3**: **Effect of Pre-Training and Extension to Channel Sharing.** Thanks! Effectiveness of pre-training depends on
tasks but we observe that it improves our performance by 11.3% in NYUv2 3-Task. We started from scratch for a fair
comparison among different methods. AdaShare can be easily extended for finding a channel sharing pattern and our
preliminary experiments on DomainNet shows encouraging results; we leave this as an interesting future work.

**R4**: **Stage-wise Training and Curriculum Learning.** We follow [55,58] and adopt a two stage training approach to
ensure the feature sharing pattern generalize to the validation dataset. We observe that the network weights learned
using one stage training is not fully optimized resulting in a drop of performance by about 15% in NYUv2 2-Task. Both
Tab. 5-main and Tab. 7-supplementary shows effectiveness of curriculum learning (improvement of 3.3% in both cases).

**R4**: **Task Relationships**–See Fig.1 and analysis at top for diverse task correlations in DomainNet. **NYU v2 Surface**
**Normals**–We use publicly available surface normals provided by [15]. **Clarity Issues**–We will fix them in final version.

**References:** [a] Peng et al, Moment Matching for Multi-Source Domain Adaptation, ICCV'19. [b] Chen et al, Exploring Shared Structures and Hierarchies for Multiple
NLP Tasks, arXiv'18. [c] Conneau et al, Very Deep Convolutional Networks for Text Classification, EACL'17.


[Meta-Review · NeurIPS 2020]

A good paper that presents a new multitask learning method. The reviewers agree that the paper is well written and the results support the main claim of the paper. The reviewers have some concerns regarding the applicability of the proposed method to other domains. It would be good if the authors can address this in the revised version of the paper.